**Knowledgebase & Database Resources**

# PomBase: a Global Core Biodata Resource—growth, collaboration, and sustainability

Kim M. Rutherford [ID] ,[1,*] Manuel Lera-Ramírez [ID] ,[2] Valerie Wood [ID] [1,*]

[1]Department of Biochemistry, University of Cambridge, Cambridge CB2 1GA, UK
[2]Department of Genetics, Evolution and Environment, University College London, London WC1E 6BT, UK

*Corresponding author: Department of Biochemistry, University of Cambridge, Sanger Building, Old Addenbrooke's Site, Tennis Court Road, Cambridge CB2 1GA, UK.
Email: kmr44@cam.ac.uk; *Corresponding author: Department of Biochemistry, University of Cambridge, Sanger Building, Old Addenbrooke's Site, Tennis Court Road, Cambridge CB2 1GA, UK. Email: vw253@cam.ac.uk

PomBase (https://www.pombase.org), the model organism database (MOD) for fission yeast, was recently awarded Global Core Biodata Resource (GCBR) status by the Global Biodata Coalition (GBC; https://globalbiodata.org/) after a rigorous selection process. In this MOD review, we present PomBase's continuing growth and improvement over the last 2 years. We describe these improvements in the context of the qualitative GCBR indicators related to scientific quality, comprehensivity, accelerating science, user stories, and collaborations with other biodata resources. This review also showcases the depth of existing connections both within the biocuration ecosystem and between PomBase and its user community.

Keywords: model organism; model organism database; MOD; fission yeast; *Schizosaccharomyces pombe*; Global BioData Coalition; Global Core Biodata Resource; standardization; FAIR; database; knowledgebase; biocuration

## Introduction

Understanding the intricate mechanisms and processes that govern the behavior of eukaryotic cells is a fundamental goal in modern biology. *Schizosaccharomyces pombe* (*S. pombe*) is one of the model organisms critical to this pursuit. Research utilizing *S. pombe* has played a pivotal role in elucidating fundamental aspects of cell cycle control (Nurse 2020), cell division (Mangione and Gould 2019), chromosome biology (Sato *et al.* 2021), epigenetic inheritance (Grewal 2023), telomere biology (Kanoh 2023), and many other core-conserved cellular processes. More recently, aging (Ohtsuka *et al.* 2023), autophagy (Alao *et al.* 2023), RNA processing (Larochelle *et al.* 2017), posttranscriptional regulation (Hernández-Elvira and Sunnerhagen 2022), autophagy (Xu and Du 2022), and mitochondrial processes (Dinh and Bonnefoy 2023) have become areas of increased focus.

PomBase (https://www.pombase.org) is the authoritative model organism database (MOD) for *S. pombe*, serving as a comprehensive knowledgebase that supports fission yeast researchers and the broader scientific community (Lock *et al.* 2020; Harris *et al.* 2022; Toda *et al.* 2023). Through detailed curation, standardization, and integration of information derived from thousands of focused experiments, it provides a repository for molecular data at the level of the gene and protein. PomBase aims to be a fully Findable, Accessible, Interoperable and Reusable (FAIR)-compliant resource (Wilkinson *et al.* 2016). In addition to a sophisticated query tool, PomBase also offers organism and domain-level overviews across multiple axes of classification including function, process, location, phenotype, human disease gene ortholog, curation, and characterization progress.

By consulting PomBase at every stage of the research cycle, the fission yeast community gains valuable insights to plan experiments, interpret experimental data, analyze results, and inform subsequent investigations. The integration of diverse data sets enables meaningful connections to be made between new and existing information, stimulating novel scientific discoveries that contribute significantly to advancements in our understanding of eukaryotic cell biology.

Since the previous update in 2022, content curated by PomBase has featured prominently in many publications including studies related to the regulation of heterochromatin spreading (Greenstein *et al.* 2022), mitotic spindle dynamics (Lera-Ramirez *et al.* 2022), identification of novel genes involved in sexual reproduction (Billmyre *et al.* 2022), and defining contractile ring components (Snider *et al.* 2022). The genome sequence and annotated features from PomBase were used for studies including the dissection of epigenetic pathways (Bao *et al.* 2022), translational control (Duncan and Mata 2022), nuclear membrane integrity (Ader *et al.* 2023), and work supporting the role of inositol pyrophosphate (IPP) as an agonist of RNA 3′-processing and transcription termination (Schwer *et al.* 2022). PomBase's curated resources have also been used to study human disease networks and mechanisms (Chesnel *et al.* 2020; Calvo *et al.* 2021; Minnis *et al.* 2021) and in the study of the conserved mechanisms affecting aging and lifespan (Romila *et al.* 2021; Uehara *et al.* 2021; Mori *et al.* 2023). Some human genes (>350) are conserved in *S. pombe* but absent from budding yeast, making *S. pombe* a powerful single-cell organism for the study of the related human gene functions. Recent examples where PomBase played a supporting role in the

characterization of genes conserved in humans but absent in budding yeast include the identification of novel factors for mRNA splicing (Anil *et al.* 2022; Selicky *et al.* 2022). PomBase extends its reach beyond fission yeast by disseminating curated knowledge to other biological databases and also by providing support for functional inferences which inform the interpretation of conserved gene functions in many other species including human (Feuermann M., Huaiyu M., Gaudet P. *et al.*, manuscript in review).

PomBase was recently awarded Global Core Biodata Resource (GCBR) status by the Global Biodata Coalition. The GCBR's mission is to define fundamental biodata resources worldwide to enable funders to better coordinate the funding, management, and growth of the research infrastructure (Anderson and Global Life Science Data Resources Working Group 2017). Resources are assessed across a range of indicators: (1) scientific focus and quality of science; (2) the community served by the resource; (3) quality of service; (4) funding, governance, and legal infrastructure; and (5) accelerating science (see https://zenodo.org/records/7468719 for a full list of indicators).

Some of the individual indicators for compliance are straightforward to quantify, implement, and evaluate. The community served by the resource can be measured by metrics including usage (via web analytics), acknowledgements, and citations. Quality of service can be assessed using percentage availability per month, response times, backup and disaster recovery, and the use of recognized standards for recording data provenance. Funding, governance and legal infrastructure criteria are largely satisfied when appropriate licensing and an active advisory board are in place. Other aspects of resource data management, maintenance, and growth are more scientifically relevant but are necessarily qualitative and are difficult to describe or evaluate because scope, specialization, data sources, data types, and data volume vary between resources. For example, it is currently difficult to establish accurately the size and content of the curatable literature corpus for a specific species or data type. The community contribution to a resource and the breadth and depth of curation coverage (curation completeness) for different data types are also difficult to assess because appropriate metrics have not been defined. The extent of collaboration and data and software sharing between resources and the value to the research community generated by improved efficiency is similarly difficult to quantify. The final GCBR evaluation category "accelerating science" is the most challenging area in which to demonstrate compliance because it involves a qualitative assessment of how effectively the data resource is meeting the needs of the scientific community via counterfactual and "accelerating science" user stories (Global Biodata Coalition 2022 Dec 21). These difficult-to-evaluate, qualitative indicators are critical for the understanding of global biodata resource contributions and sustainability efforts.

For PomBase, with a staff of only 2–3.5 full-time equivalent (FTE), it is essential that resources are deployed efficiently and outputs produced by other groups are not duplicated. Resource sharing strategies deployed at PomBase range from the development of reusable extendible open-source software, reuse of software and tools developed by others, data sharing (inwards and outwards) and the co-development of standards and quality control systems. All these strategies improve efficiency while continuing to meet evolving user requirements and contributing to resource sustainability efforts across the bioresource ecosystem.

In this database update, we present PomBase's continuing growth and improvement since the previous resource update in 2022 (Harris *et al.* 2022). Improvements are discussed in the context of the qualitative GCBR indicators related to scientific quality, evaluation of comprehensivity, accelerating science, and the connections, collaborations, and data sharing with other biodata resources.

## Assessing scientific quality: growth, completeness, and coverage

In the biocuration field, resource productivity can be simplistically measured by curated data increase. In PomBase, experimentally supported curated statements or annotations, sourced from the primary literature, show a consistent year-on-year growth, increasing from 287,000 in January 2022 to 392,000 as of November 2023 (Fig. 1a). For a knowledgebase, curation growth can include both novel data and duplicate statements from different experiments. Recording redundant observations of different provenance and from different methods is necessary because it increases confidence in the result or broader hypothesis. However, metrics evaluating knowledge accumulation based solely on data growth do not provide information about the knowledge coverage (or completeness) in a specific domain. Users need to be confident that knowledge coverage is adequate for data analyses or interpretation of results making it a pivotal aspect of scientific quality. Knowledge coverage grows more slowly than annotation count because only novel, previously uncurated observations contribute to its increase. Assessing knowledge completeness is challenging because an absence of coverage can be due to either uncurated publications or an absence of published data. Methods used to assess the knowledge coverage and target knowledge gaps at PomBase are discussed in the next section.

### Literature curation coverage and contents

Once all "curatable" publications are identified, the percentage of fully curated publications provides an additional metric to assess curation completeness. In some domains and for some model species, the volume of literature makes it difficult to establish even the approximate number of "curatable" publications. For PomBase, identifying domain-specific literature is relatively straightforward by retrieving publications from PubMed that mention "fission yeast" or "S. pombe" (14,155 publications, November 2023). The manual triage process examining title and abstract identifies incorrectly retrieved publications, for example, when fission yeast is mentioned in the abstract but is not the experimental system used (3199 publications). Publications are then classified as either "uncuratable" or "curatable." Uncuratable publications are in scope for PomBase by topic but have no obvious gene-specific experimental data; these publications are classified into subtopics [for example, biotechnology or toxicity studies (89), theoretical modeling (111), methods or reagents (967), wild-type features (301), or review (1348)]. This triage process classifies publications within PomBase and also effectively creates a training data set for machine learning (ML) publication classifiers. The remaining "curatable" publications are expected to contain experimental data (6724). The difference between the number of "curatable" (6724) and "curated" (4440) publications (2284) provides an alternative measure of curation completion (66% for PomBase).

### Using high-level overviews or "slims" to assess curation coverage

To describe biological observations consistently, most curation activities make extensive use of defined domain-specific concepts

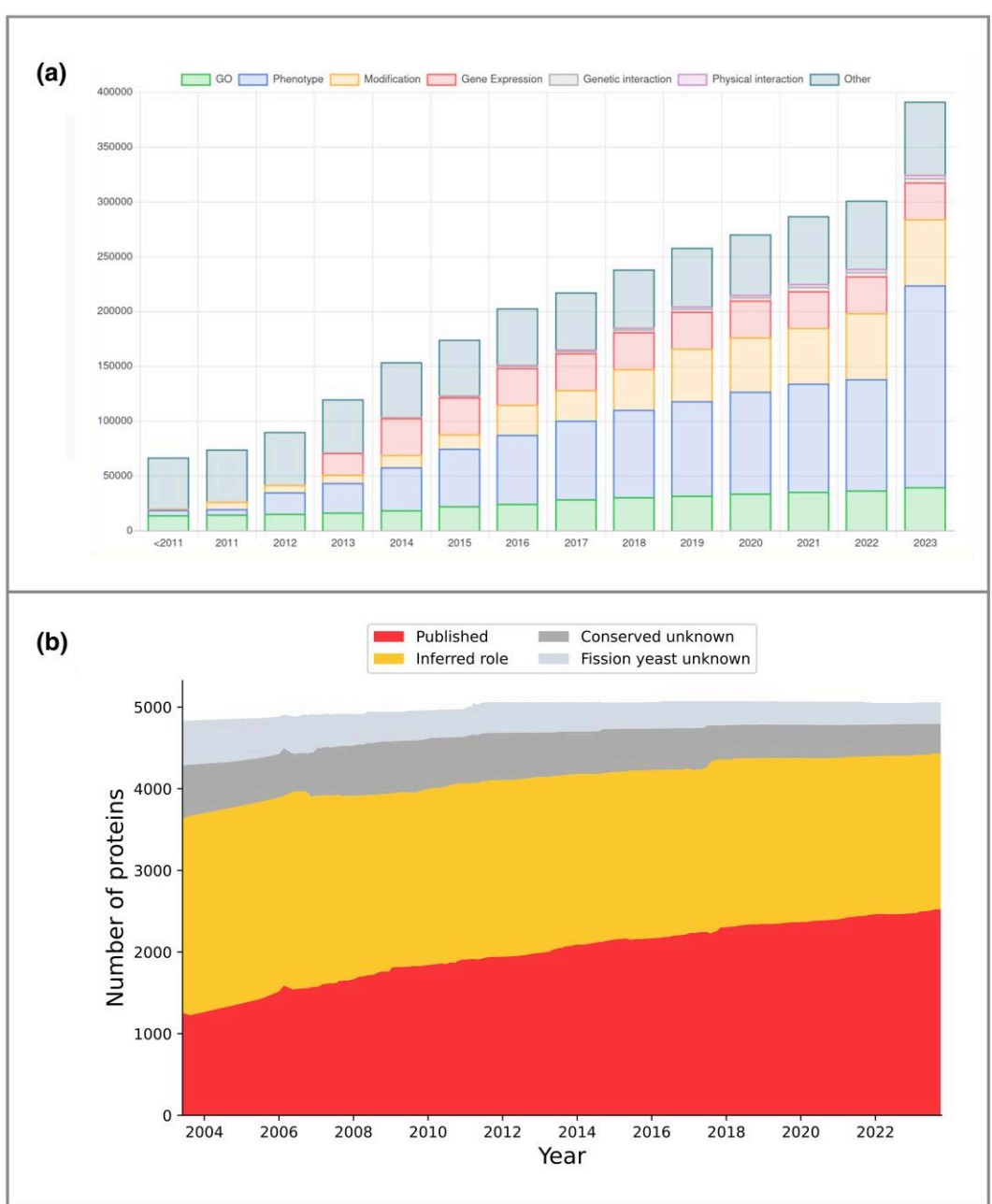

**Fig. 1.** Metrics to assess curation growth and coverage. a) Experimentally supported curated statements or annotations, sourced from the primary literature, show a consistent year-on-year growth (https://curation.pombase.org/pombe/stats/annotation). Major data types collected by PomBase are shown. b) Classifying proteins as functionally "known" or "unknown" is 1 way to describe the completeness of both curation and scientific knowledge (https://www.pombase.org/status/gene-characterisation). Here, we classify protein-coding genes as "known" or "unknown" based on GO biological process assignment to biologically meaningful grouping classes and show changes over time. We separate known into (1) "known from an experiment in *S. pombe*" (red / bottom band), where the process is completely or partially characterized in low-throughput experiments in *S. pombe*, and (2) "known from an inference from another species" (yellow / band second to bottom). We separate "unknown" into (1) "conserved unknown" (dark gray / band second from top), present in some species outside the *Schizosaccharomyces* clade, and (2) "species-specific unknown" (light gray / top band), restricted to the fission yeast clade.

("terms" or "classes") from ontologies approved by the Open Biological and Biomedical Ontology (OBO) Foundry (Jackson *et al.* 2021). Ontologies are directed graphs where the root node defines the most general classes [for example, Gene Ontology (GO) "biological process"], while more specific terms ("subclasses") describe increasingly specific concepts (i.e. "cellular metabolic process," "DNA metabolic process," "DNA replication") (Ashburner *et al.* 2000). One way to provide an overview of knowledge coverage is via an ontology "slim set," a set of informative "high-level" biologically meaningful grouping classes (Lock *et al.* 2018). We use the biological process aspect of the GO (one of the most widely used bio-ontologies) to illustrate ontology overviews (Gene Ontology Consortium *et al.* 2023).

The "slim set" for the GO biological process ontology relevant to a single-celled species includes terms such as "DNA replication," "transmembrane transport," "mitochondrion organization," and "lipid metabolic process" that represent biological modules of interest to groups of researchers with a specific focus. PomBase provides slim sets, accessible prominently from the PomBase front page, for each of the 3 GO aspects (molecular function,

biological process, and cellular component), for disease associations of known human orthologs, and most recently for phenotypes (https://www.pombase.org/browse-curation/fypo-slim). Lists of genes annotated to each slim term (e.g. "DNA replication") and its descendants/subclasses can be browsed, downloaded, or used in queries. Critically, to allow knowledge coverage to be assessed, each "slim" page also indicates the number of gene products not covered by the slim set.

There are 2 reasons that genes may not be assigned to a GO slim category: (1) the annotated terms are not currently covered by the slim set, or (2) the gene products are currently *unannotated*. Gene products may be *unannotated* because they are *unknown* (*uncharacterized* or *unpublished*) or because they are *curatable* (i.e. known) but have yet to be curated.

PomBase prioritizes broad GO annotation coverage and accuracy alongside specificity. We consider general and correct curation over the entire corpus to be more critical for genome-wide investigations than annotation completeness in only the areas of biology that are intensively studied in fission yeast. However, both breadth and depth (specificity) of annotation completeness are the long-term aims. GO breadth of coverage is improved by identifying and targeting knowledge gaps in the uncurated (i.e. the *unannotated* but *curatable*) portion of the GO slim sets. We believe that our GO process coverage at the slim level (breadth) is close to completion (PomBase GO biological process slim; https://www.pombase.org/browse-curation/fission-yeast-bp-go-slim-terms) and that we are making good progress with specificity (based on literature completion metrics above).

Once complete coverage of a slim set is established using all available published or inferred information, increased coverage will only be derived from newly published gene characterizations (in either the species of interest or inferred from another species) (Wood *et al.* 2019) or inferences from newly identified distant orthologs of previously characterized genes, for example using AlphaFold (Monzon *et al.* 2022). However, at PomBase (and likely other MODs), much of the valuable experimental detail, including connections between gene products (i.e. protein kinase or transcription factor targets) and much of the reproduced experimental data that will provide increased confidence in curated statements, remains to be captured. No standard metrics exist to assess breadth or depth of curation, and it is therefore difficult to establish how complete knowledge capture is in any domain for available published results.

## Unknown (uncharacterized) proteins

Proteins without a known function have become an area of increased concern in recent years, following observations that gene characterization rates have stagnated and a large proportion of proteins remain unstudied or understudied even in model organisms (Sinha *et al.* 2018; Wood *et al.* 2019; Kustatscher *et al.* 2022; Rocha *et al.* 2023). Classifying proteins as functionally known or unknown is an additional way to describe the completeness of both curation and scientific knowledge (Fig. 1b).

There are multiple ways to classify whether a gene is functionally characterized, and the concept of "characterization" is clearly a continuum. However, as an initial step in defining "unknowns," we devised a simple binary classification, based on biological processes at the level of the GO slim set (see above) (Wood *et al.* 2019). This classification requires a meaningful biological process assignment (i.e. program or pathway) rather than a precise molecular function (activity). We select meaningful biological process terms as used on the slim set (i.e. DNA replication, translation, chromosome segregation, cytokinesis) and omit general terms

(i.e. metabolism, cellular process, and responses to various chemicals). The rationale for this process-focused approach is that (1) the same molecular functions are often reused in many different cellular contexts, (2) most researchers focus on biological programs rather than specific activities, and (3) many proteins are members of protein complexes that are relatively well-studied at the level of the biological process, but the precise molecular function of each subunit is not known or difficult to describe. For example, subunits of well-studied complexes [e.g. anaphase-promoting complex (APC) or ESCRT III complex] will often have no assigned molecular function beyond "protein binding" but are well-studied at the process level. We reason that knowledge of a specific, but well-evidenced biological, process is usually sufficient to bring a partially characterized gene to the attention of the relevant researchers for more detailed investigation. We further separate these functionally classified proteins depending on whether the biological process is derived directly from studies in fission yeast or is inferred from orthologs in other species (Fig. 1b). In addition, we classify the "functionally unknown" proteins based on whether they are conserved outside the *Schizosaccharomyces* clade or if they are currently identified as *Schizosaccharomyces* specific. This classification allows us to identify universally conserved, or eukaryotically conserved, unknown proteins and provide inventories of priority unstudied genes or poorly characterized genes. This list of unstudied genes is updated daily (https://www.pombase.org/status/priority-unstudied-genes).

The PomBase unknown protein inventory was first published in 2019. Since then, the number of unstudied proteins using this metric has decreased from 693 to 618. We periodically review unknown proteins for available functional data from other species.

## Collaboration: software and data sharing
### Reuse of external software to enhance PomBase

When PomBase requires new tools and visualizations for curated or imported data, we evaluate and, whenever possible, reuse existing software rather than developing new tools in house.

Since 2018, the PomBase gene pages have featured the JBrowse genome browser widget in the summary section (Buels *et al.* 2016). The genome sequence and the 316 accompanying data tracks hosted in the full JBrowse instance (https://pombase.org/jbrowse/) are of general interest to most users and of special interest to the gene expression and heterochromatin communities. However, we found from user feedback that the genome context is not always the most biologically relevant summary display on gene pages (for example, structural biologists or cell biologists). The new gene page summary section allows users to set their preferred view to JBrowse or one of several new views described below (Fig. 2a).

### *Protein feature viewer*

PomBase phenotype annotation captures the results of genetic experiments by linking a phenotypic observation to the genotype of a mutant strain and to the experimental conditions. Phenotypes are the most commonly annotated data type in PomBase, with an increase from 99,300 annotations in January 2022 to over 185,000 as of November 2023. This genotype-to-phenotype data include 6442 alleles encoding protein variants (substitutions and partial deletions) used for over 28,700 phenotype annotations. The continual growth of phenotype data and the simultaneous growth in curated protein modifications (to over 60,000) requires new tools to enable users to view potentially hundreds of data points from different data types simultaneously for any single gene/protein. To make this possible, we have implemented a

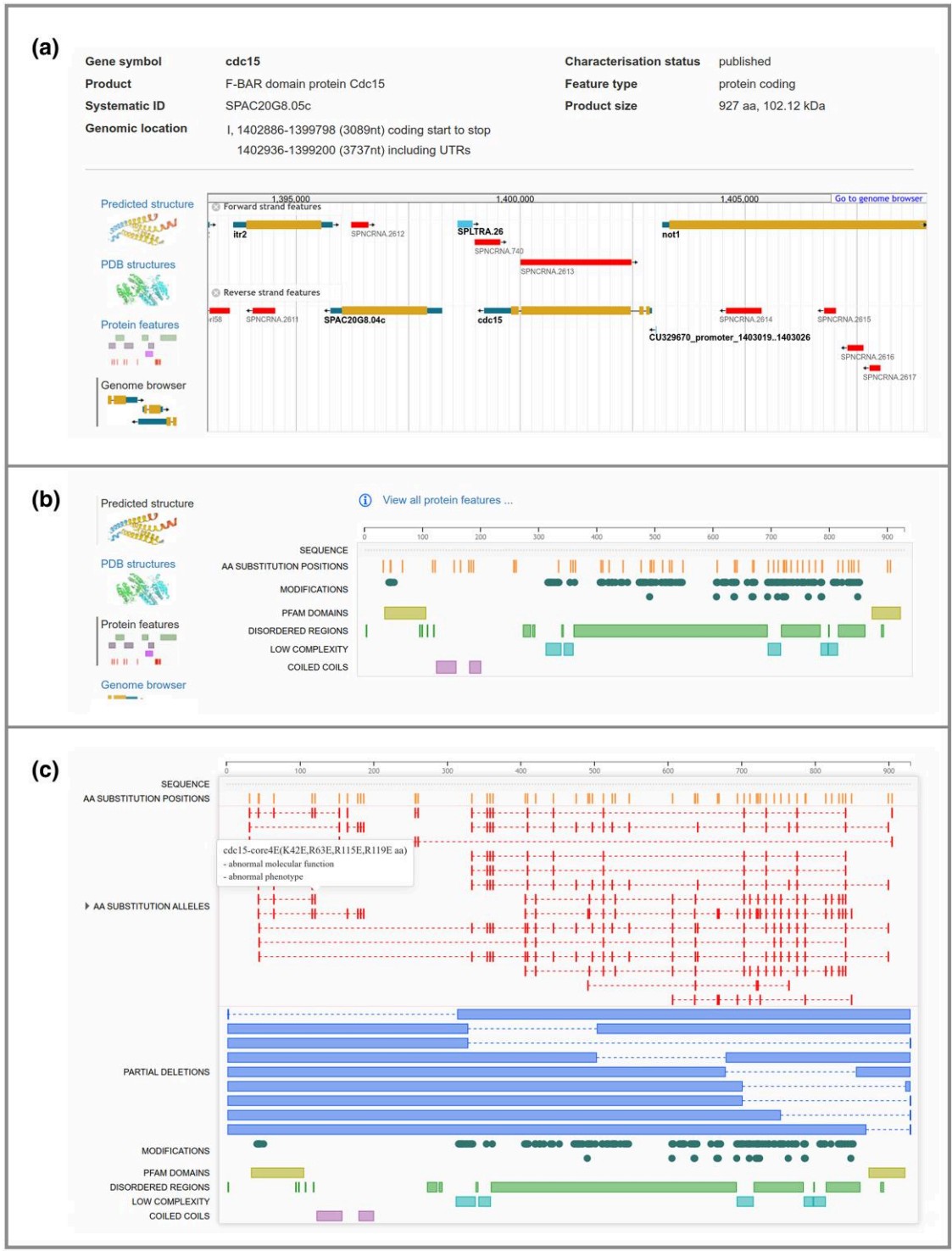

**Fig. 2.** New gene page summary and protein feature views. a) The gene page summary section has been redefined to allow users to set their preference to the existing JBrowse (genome browser) view or one of several new views. b) The new protein feature summary view uses the RCSB 1D Feature Viewer to display protein features (i.e. mutated residues, modifications, and structural features), illustrated here using the Cdc15 F-Bar membrane adaptor (https://pombase.org/gene/SPAC20G8.05c). c) The new protein feature "full view" also uses the RCSB 1D Feature Viewer and is accessible via a link from the summary view. This view provides a comprehensive display of engineered amino acid mutations (substitutions and deletions) linked to phenotype information. Mouse-over provides feature details. Illustrated here using Cdc15 protein F-Bar membrane adaptor (https://pombase.org/gene_protein_features/SPAC20G8.05c).

protein feature summary using the Research Collaboratory for Structural Bioinformatics (RCSB) 1D Feature Viewer (Segura *et al.* 2021). The new protein feature summary view displays amino acid mutagenesis and modifications in the context of the protein sequence (Fig. 2b). The linked full protein feature view includes all amino acid mutations for single locus genotypes including multiple amino acid substitution alleles and serial deletion alleles (Fig. 2c).

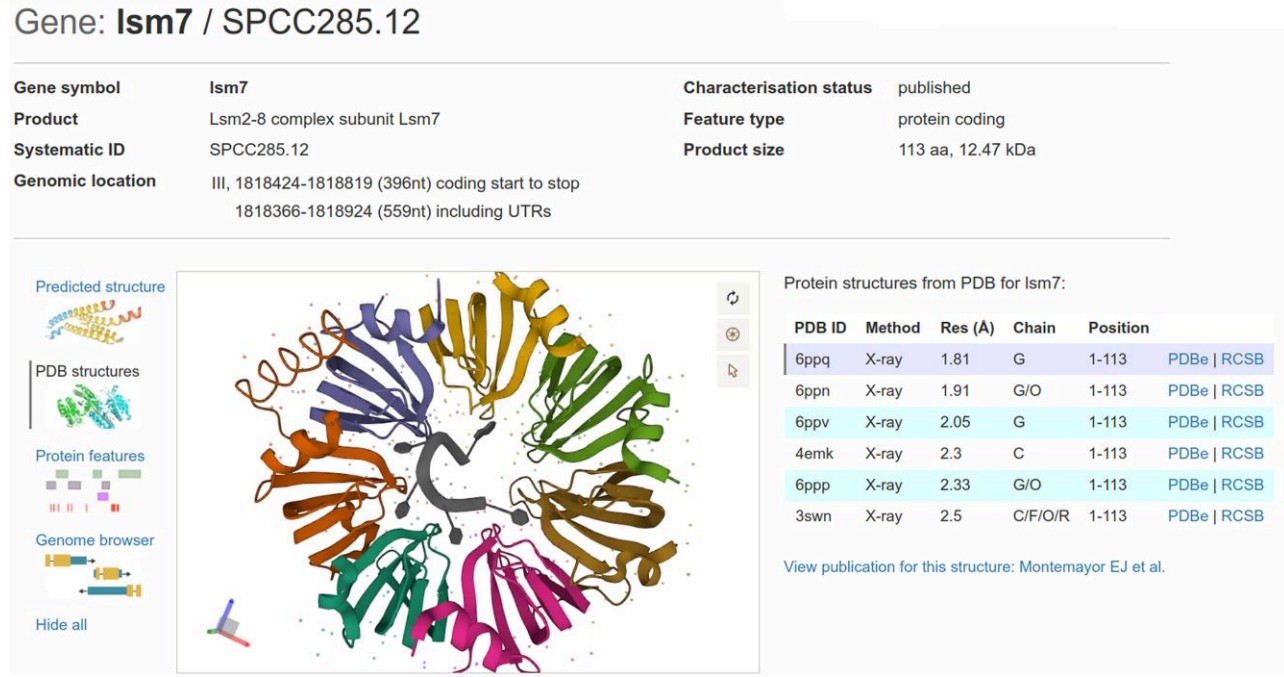

**Fig. 3.** New structure views. Mol* (/molstar/), a web-based open-source toolkit developed by RCSB-PDB, is used to display protein structures from the PDB and AlphaFold. Illustrated here using the Lsm7 protein PDB entry as part of the Lsm2-8 complex (https://www.pombase.org/gene/SPCC285.12).

### Protein structure views using Mol*

We recently imported and displayed *S. pombe* experimental protein structures from Protein Data Bank (PDB) (484 structures for >400 proteins) and protein structures predicted by AlphaFold (5124 proteins, database version 2022 November 1) into PomBase (see also data reuse section below) (PDBe-KB consortium 2020; Varadi *et al.* 2022). We display these structures using Mol* (/molstar/), an open-source JavaScript library developed by RCSB-PDB (Sehnal *et al.* 2021). Mol* provides high-performance graphics and data handling allowing users to visualize and interrogate protein structures. We configured Mol* to provide informative views and metadata based on consultation with structural biologists, illustrated here using the Lsm7 protein (Fig. 3). We provide the same protein structure view on the publication pages for any research publication containing an experimental protein structure. In the future, we will link mutated residues in protein feature view (Fig. 2c), to the corresponding residues in the protein structure view.

### Viewing reactions with the Rhea widget

A multiyear collaboration between the GO consortium (GOC) and the Rhea reaction database has mapped more than 4500 GO catalytic activities to Rhea reactions (Gene Ontology Consortium 2021; Bansal *et al.* 2022). We were able to take advantage of this valuable resource and the reaction diagram widget from Rhea, to display chemical equations alongside the associated activities on 794 GO molecular function pages (Fig. 4). These reaction widgets are also available via the gene pages by mouse-over any associated molecular function.

## Reuse of PomBase software by other resources

### PomBase website Code

The PomBase website code provides intuitive displays, a versatile query system, and visualization tools (described above) (Lock *et al.* 2019; Harris *et al.* 2022). The website system is supported by a data import and processing pipeline that allows daily updates and a curation database using the Chado schema (Mungall *et al.* 2007).

The website code is designed to be low maintenance and for long-term sustainability. It is easily extendable to incorporate novel data types and new visualizations enabling rapid reuse by other communities including emerging model species.

Since the 2022 MOD resource update publication (Harris *et al.* 2022), we have made many extensions to the website code. In addition to the protein structure and protein feature views described above, we have improved the Advanced Query tool to enable querying by transcript number, RNA length (spliced and unspliced), and presence of coiled-coils or disordered regions (for protein-coding genes).

We previously reported the implementation of JaponicusDB (https://www.japonicusdb.org/) for the emerging model fission yeast *Schizosaccharomyces japonicus* as part of a 2-month project using the PomBase website and database system (Rutherford *et al.* 2022). This is, to our knowledge, the first MOD maintained entirely by the research community with no dedicated professional biocuration effort. The JaponicusDB user base is relatively small (approximately 2000 unique users per year compared to 45,000 for PomBase), and the active community is substantially smaller. This model is effective for this small but highly motivated community because basic content can be provided and managed by the users with no manual intervention by PomBase staff. There is no development overhead since all new PomBase features (as described above) are automatically deployed nightly for JaponicusDB.

### Canto curation tool reuse

The online curation tool Canto (https://curation.pombase.org/) is developed and maintained by the PomBase team to support curation by professional curators and publication authors (Rutherford *et al.* 2014). Canto supports genotype management and ontology-based curation (including GO, phenotypes, and protein modifications) with associated metadata, provenance, and

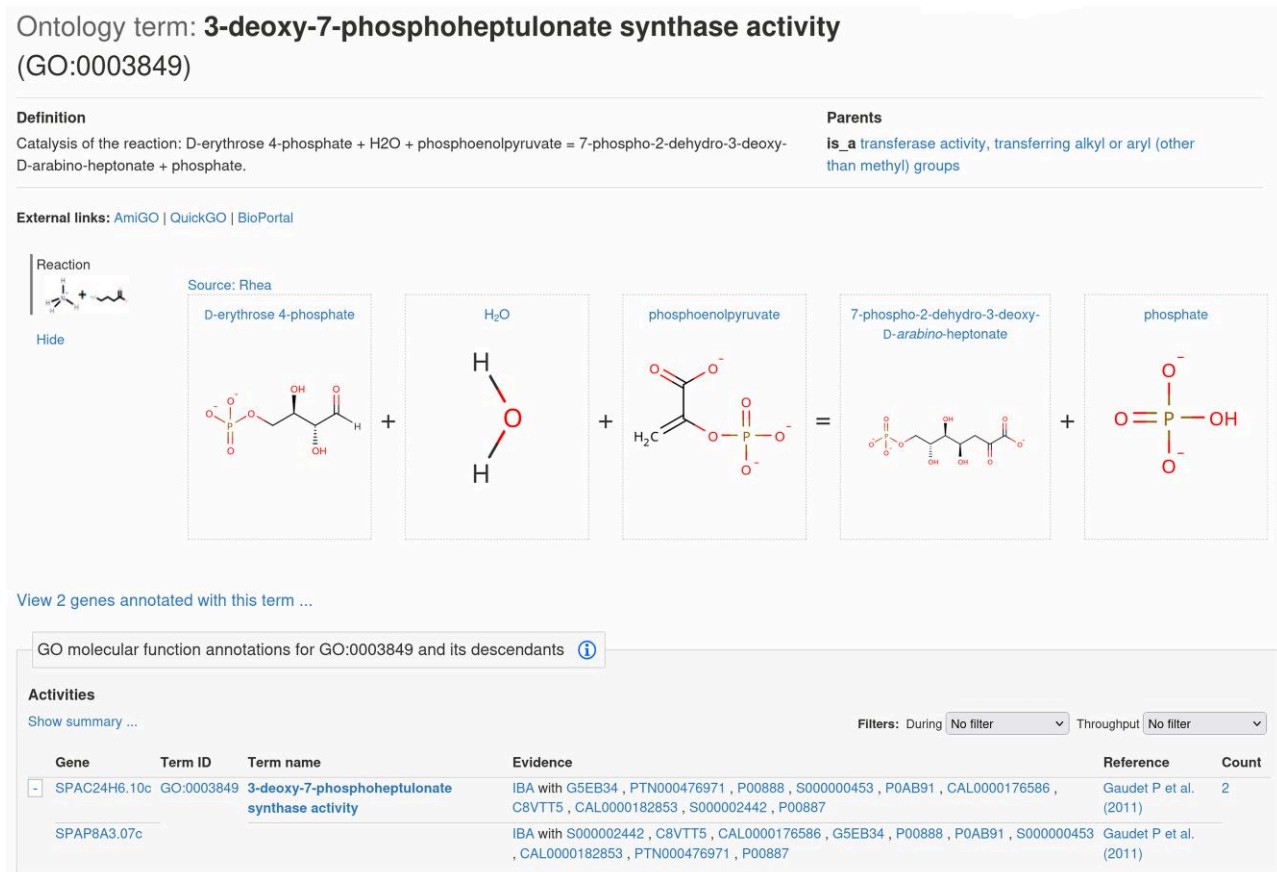

**Fig. 4.** The Rhea reaction widget. The widget provided by the Rhea reaction database makes use of the GO to Rhea mappings to display reactions for all annotated GO activities.

annotation extensions (Huntley *et al.* 2014). Canto is easily extendable to incorporate additional ontology-based annotation types with associated metadata. Canto has now been deployed for several other communities, including PHI-base for pathogen–host interaction phenotypes (Urban *et al.* 2017; Cuzick *et al.* 2023), FlyBase for phenotype annotation of *Drosophila melanogaster*, and the *S. japonicus* MOD JaponicusDB as described above.

## Data dissemination from PomBase to other resources

PomBase disseminates curated data sets to a wide range of biological repositories and knowledgebases (Fig. 5). We provide over 27,000 experimentally supported, manually curated annotations to the GO database (Gene Ontology Consortium *et al.* 2023), accessible via the AmiGO browser (Carbon *et al.* 2009). These data are further disseminated by the GOC to the UniProtKB GOA database (Huntley *et al.* 2015), accessible via the QuickGO browser (Binns *et al.* 2009), and disseminated by GO or GOA to FungiDB, NCBI, and Ensembl Genomes.

The experimental knowledge captured in GO statements is subsequently used to inform unstudied species and to fill annotation gaps in well-studied species by annotation transfer. This is probably the most important activity of the MODs after supporting their immediate research community (Oliver *et al.* 2016). PomBase is one of the richest sources of experimental cell-level curation for transfer to orthologs in other species via the GO PAN-GO project (Gaudet *et al.* 2011). Via the PAN-GO pipeline, we provide experimental support for 565,172 inferred GO annotations

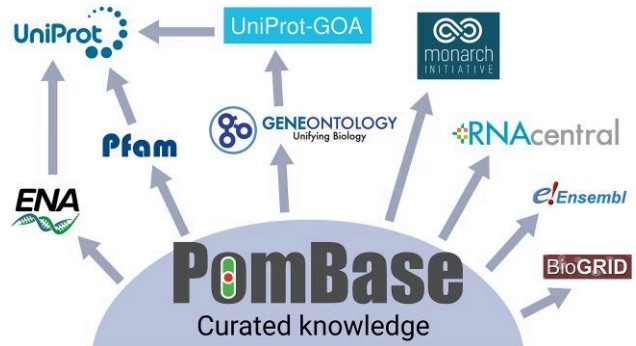

**Fig. 5.** Major knowledge dissemination pathways from PomBase. PomBase provides regular updates of curated data to major bioresources and repositories. We export manually curated gene structures to the ENA. ENA gene structures are used to provide the UniProtKB protein sequence (and subsequently used by InterPro resources and AlphaFold). We submit protein families and protein family updates to Pfam. As part of the GOC, we export functional annotation to the GO repository which is further disseminated to GOA and UniProtKB (and to NCBI and FungiDB, not shown). Phenotype data are exported to the Monarch Initiative knowledgebase. RNA features and GO curation are exported to RNACentral. The genome and nucleotide sequences and features are imported by Ensembl Genomes. Protein and genetic interaction data are submitted to BioGRID.

across ~150 key species increasing to 697,000 for all inference methods (data from QuickGO, 2023 November 20). Most of the propagated annotations inform the biology of unstudied or underannotated fungi. However, because, *S. pombe* proteins are broadly

conserved among eukaryotes (over 70% have human orthologs), *S. pombe* GO curation is also used to support over 6800 inferred human annotations (increased from 3863 in 2019) (Feuermann M., Huaiyu M., Gaudet P. *et al.*, manuscript in review).

We have recently submitted 100,000 *S. pombe* phenotype annotations to the Monarch Initiative, an integrative data and analytic platform connecting phenotypes to genotypes across species (Shefchek *et al.* 2020). This will make fission yeast data available to compare phenotypes between species and to explore the effects of directed mutations of conserved protein residues. These data are further disseminated by Monarch to other resources, including the Critical Path Institute (https://c-path.org/), to support collaborations that accelerate drug development.

PomBase exports ncRNA data and annotation to the RNACentral RNA repository (RNAcentral Consortium 2021). Export files are updated daily and ingested by RNACentral at each release ~3 times annually. The number of annotated fission yeast noncoding RNAs (ncRNAs) has increased from 318 to 7518 since 2022. Some ncRNA features have been merged, some unsupported ncRNAs have been deleted, and several recently published ncRNAs have been annotated with GO terms and phenotypes.

PomBase has collaborated with BioGRID since 2013 to curate physical and genetic interactions (Oughtred *et al.* 2021). Interaction data curated by BioGRID are refreshed in PomBase after each BioGRID release, and reciprocally, interaction data curated from small-scale experiments by PomBase are periodically submitted to BioGRID. We recently extended the data model for genetic interactions at PomBase to associate the interactions with genotypes rather than genes and to connect them with the phenotypes they enhance or suppress. This change will help geneticists interpret interactions more quickly and will improve consistency between phenotype and genetic interaction annotations. After these improvements, the PomBase data processing pipeline continues to support the export of the gene-to-gene connections to BioGRID.

As the database authority with responsibility for genome feature coordinate changes, the accuracy of *S. pombe* gene structures at NCBI, Ensembl Genomes, and FungiDB and their derived protein sequences in UniProtKB, InterPro, and AlphaFold is dependent on PomBase. We have also improved the submission process to support European Nucleotide Archive (ENA) updates with each PomBase release. This streamlined update procedure will ensure that all resources have rapid access to accurate gene structures. Finally, we also submit novel protein families and updates for domains of unknown function (DUFs) to the Pfam protein family database.

## Data ingested into PomBase from other resources

During each nightly update, PomBase imports data from many other sources. Genetic and physical interaction data are imported from BioGRID as part of a reciprocal arrangement (export is described above). Manual GO annotation is supplemented by phylogenetically inferred annotations provided by the GOC PAINT pipeline (Gaudet *et al.* 2011). In addition, computationally inferred GO annotation is imported from the GOA database (Huntley *et al.* 2015). These annotations are provided by the InterPro2GO mappings maintained by InterPro (Burge *et al.* 2012), by Rhea2GO mappings maintained by the GOC and Rhea, and by the UniProtKB mappings derived from keywords, pathways, subcellular locations, and the UniProtKB rule-based system (UniProtKB-KW, UniPathway, UniProtKB-SubCell, and UniRule, respectively) (Bansal *et al.* 2022; UniProt Consortium 2023).

Protein domains and families are imported from InterPro (Paysan-Lafosse *et al.* 2023). Known protein structures are imported from PDB and predicted structures from the AlphaFold database (wwPDB consortium 2019; Varadi *et al.* 2022). All article details (author, title, abstract, etc.) are imported from PubMed. Inferred molecular pathways are imported from the Kyoto Encyclopedia of Genes and Genomes (KEGG) database (Kanehisa *et al.* 2023).

We maintain manual inventories of orthologs between *S. pombe* and *Saccharomyces cerevisiae* and between *S. pombe* and human based on integrated predictions and divergent orthologs derived from experiments (i.e. co-conservation of protein complex subunits). For these orthologs, the HUGO Gene Nomenclature Committee (HGNC) and Saccharomyces Genome Database (SGD) symbols and descriptions are imported to display on the gene pages (Seal *et al.* 2023; Wong *et al.* 2023). For orthologs in other species, we provide links to the DRSC Integrative Ortholog Prediction Tool (DIOPT) which integrates ortholog predictions from over 20 providers (Hu *et al.* 2011). We also map Monarch disease ontology (Mondo) terms from human disease-associated proteins to orthologous pombe proteins (Vasilevsky et al. 2022). Over 3500 fission yeast proteins have human orthologs, 1513 of which are mapped to disease genes (increased from 1401 in August 2021). Disease associations allow users to retrieve all genes implicated in specific disease types (i.e. premature aging syndromes or metabolic disorders) using the PomBase Advanced Search tool. This resource has been used to classify diseases caused by interactors of btn1 (the ortholog of the human gene causative for Batten disease) (Minnis *et al.* 2021) and identify mitochondrial diseases associated with 65 nonessential fission yeast mutants that fail to proliferate in low glucose (Mori *et al.* 2023).

## Collaboration to develop global standards

The OBO Foundry ontologies used by PomBase include GO, the Sequence Ontology (SO) (Eilbeck *et al.* 2005), the Protein Ontology (PRO) (Natale *et al.* 2017), the Mondo disease ontology (Vasilevsky et al. 2022), and the protein modification ontology (PSI-MOD) (Montecchi-Palazzi *et al.* 2008). We continue to contribute new classes, corrections, and other revisions to these ontologies and are one of the largest contributors to GO ontology development. PomBase curators develop and maintain the Fission Yeast Phenotype Ontology (FYPO) (Harris *et al.* 2013). Since the previous database publication in 2022, FYPO has grown from ~7500 terms to over 8,100 terms, and the number of phenotype annotations has increased from ~99,000 to 184,000 as of November 2023.

FYPO makes use of the Phenotype And Trait Ontology (PATO), the Chemicals of Biological Interest ontology (ChEBI), the Relations Ontology (RO), and GO to relate and logically define phenotype terms (Hastings *et al.* 2016; Gkoutos *et al.* 2018; Gene Ontology Consortium *et al.* 2023; Mungall *et al.* 2023). FYPO development uses the Ontology Development Kit (ODK; Matentzoglu *et al.* 2022), which provides a release pipeline that seamlessly incorporates ontology reasoning, continuous integration checks, and generation of production files. We participate in the Unified Phenotype Ontology (uPheno) effort to integrate multiple phenotype ontologies into a unified cross-species ontology (Matentzoglu *et al.* 2019).

## Partnerships improving curated content
### *Collaborative publishing via microPublications*

MicroPublications provide a mechanism to describe brief, novel findings, negative and/or reproduced results, and results which

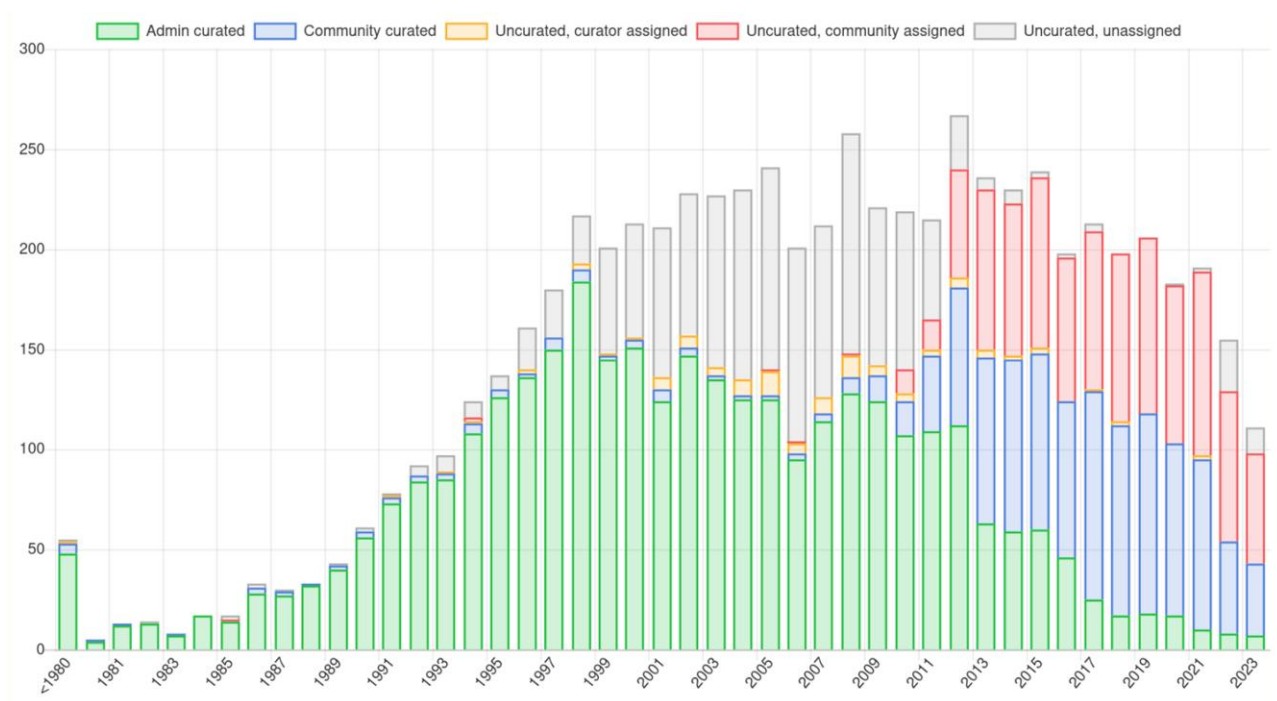

**Fig. 6.** Current low-throughput curatable publication status by year published. The current curation status of the low-throughput publications identified as containing gene-specific experimental data (i.e. curatable) classified by publication date as of November 2023. Five categories are displayed: (1) admin curated (green): papers curated exclusively by PomBase staff who concentrate on legacy publications; (2) community curated (blue): curated papers with annotations contributed by authors (these are largely restricted to papers published since 2012 when Canto was launched); (3) uncurated, curator assigned: publications assigned to a PomBase curator but not yet curated; (4) uncurated, community assigned (red): assigned to a community member but not yet curated; and (5) uncurated, unassigned: not yet assigned for curation.

may lack a broader scientific narrative and would usually remain unpublished (Raciti *et al.* 2018). PomBase has been a partner of the *microPublication* journal since 2020, to contribute to the editorial process and ensure compliance with data standards. The number of S. *pombe* articles published by this route has increased from 6 in 2021 to 11 in 2023 (year to November), covering topics from biochemistry, phenotypes, methods, software, and expression.

### Community collaboration by author-led curation

The most extensive collaborative project at PomBase is the long-term interaction with the fission yeast community via our community curation project. Community curation actively engages authors in building FAIR-shared biological knowledge (Wilkinson *et al.* 2016; Lock *et al.* 2020).

To date, 440 authors have curated 1,124publications generating over 23,000 standardized, evidenced, and provenanced annotations from directed gene-specific hypothesis-driven experiments (i.e. excluding high-throughput data sets). The current curation status of publications by year published is shown in Fig. 6. This ongoing collaboration with authors improves curation quality because co-curation by biological domain experts (authors) and ontology experts (biocurators) usually includes a dialogue to refine and extend annotation. Authors benefit from participation in the curation process in multiple ways including increased publication visibility, satisfying funders' dissemination criteria, and a deeper understanding of curated data for reuse purposes.

### Quality control contributions

As science progresses, new interpretations often replace existing incorrect or incomplete explanations. Often historical experiments are not replicated and are gradually revealed to be over-interpreted by newer research, although negative results or corrections are rarely explicitly published (Wood *et al.* 2020). At PomBase, curators routinely review existing annotations (experimental and inferred) for consistency and accuracy when curating newer publications. As a result of this incremental checking, PomBase provides one of the largest contributions to the correction of manual and electronically inferred GO annotations (Wood *et al.* 2020; Feuermann M., Huaiyu M., Gaudet P. *et al.*, manuscript in review). Evaluating data quality and removing outdated assertions is critical for basic research scientists but also because curated data are used as training data for function prediction, large language models (LLMs), and other ML applications. We have also developed a generic pipeline to validate and correct the protein residue positions referred to by alleles and protein modifications (Lera-Ramirez, manuscript in preparation).

Similar to many other aspects of biocuration discussed above, the value of identifying and fixing errors is difficult to quantify and even more difficult to attribute.

## Conclusions

### The interconnectivity of the biological data resource ecosystem

Biological databases may give the appearance of being disparate and autonomous resources. In reality, resources are deeply interdependent and routinely cooperate to establish standards and to exchange curated data, data models, software tools, ontologies, quality control pipelines, and data formats. Model organism resource staff frequently interact with other bioresources via GitHub, e-mail, Slack, and regular online, in-person consortia,

and International Society of Biocuration (ISB; https://www.biocuration.org/) meetings.

These—frequently organic—collaborative efforts reduce duplication of work for both curators and developers and ensure that resources can focus on other important tasks specific to their respective communities. The MODs collaborate on many large-scale projects such as the GOC (Gene Ontology Consortium *et al.* 2023) and Alliance of Genome Resources (Alliance of Genome Resources Consortium 2020) and also on many smaller projects such as the automation of gene summaries (Kishore *et al.* 2020) and collaborative quality control using shared annotation (Wood *et al.* 2020). Here, we have presented some of the PomBase features made possible by this, often informal, collaborative network in the context of the GCBR indicators related to scientific quality, collaboration, sustainability, and impact. We have described data sharing both inwards and outwards and contributions to global standards and quality control. We have provided a progress update for the most ambitious collaborative project undertaken by PomBase: the multicontributor community curation project empowering authors to FAIRify their data and knowledge derived from in-depth hypothesis-driven research.

## Difficulties defining metrics for impact

It is challenging for a knowledgebase to evaluate how it contributes to scientific impact. Some aspects of impact can be derived from curation activities related to setting and promoting standards as described above. However, it is also important to demonstrate how the resource adds value for its users by integrating knowledge in a way that is not possible for individual researchers or research groups. A recommended way to demonstrate impact is via "the counterfactual" (i.e. indicating the consequences for the biodata resource ecosystem if the resource ceased to exist). Powerful testimonials and "use cases" from user surveys (Wood *et al.* 2017) and letters of support are the most impactful way to demonstrate counterfactual arguments by explaining in detail how users engage with and depend upon biological resources. However, the qualitative nature of these outputs makes them difficult to evaluate, and the provision for including such documents in grant applications is limited.

Another difficulty is that resources such as PomBase are often not explicitly cited by users because their use is constant and incremental (i.e. regularly checking gene-specific facts and planning the next experiment). The many ways in which integrated data improve efficiency in basic biology by helping to formulate hypotheses and interpret results often do not align with citability. Despite this, we can track via citations some of the innovative ways in which key scientific investigations have used PomBase (a current selection is provided in the introduction).

It was recently proposed that knowledgebases should be weighted differently by funding bodies and that strong innovation is not necessary (Karp 2022). Knowledgebases are impactful not through novelty but because they are critical to scientific innovation while also reducing scientific costs. Across the resource infrastructure, cost savings of up to 20 times the investment are identified for bioresources (https://www.ebi.ac.uk/about/our-impact/impact-report-2021). It is clear from our community surveys that knowledgebases are vitally important as a way to connect research communities and support discovery and that without them much research would be difficult or impossible. However, demonstrating the complete range of outputs and impacts of these resources will remain a challenging task.

## Future directions
### Support for ML, artificial intelligence, and rule-based approaches

Detailed and accurate factual statements are required to train artificial intelligence (AI) and ML models. To support the emerging AI/ML field, we provide training data sets for the identification of curatable publications and data types (genotypes, phenotypes) that are used by various in-progress ML projects.

We recently revised nomenclature guidelines to make biological entities (genes, proteins, alleles) more machine-readable to improve automated named entity recognition by publishers (Lera-Ramírez *et al.* 2023). As a proof of principle, Genetics, G3, and microPublications now provide deep links from publications into PomBase for gene mentions (e.g. cdc15), by using a lexicon of PomBase entities. To reduce manual input steps in curation, we plan to identify named entities (genes, proteins, and alleles) in publications to preload the Canto curation tool (Rutherford *et al.* 2014).

We are actively working with AI and ML communities as the "human-in-the-loop" to support the application of machine-based methods. Work to date in this area includes evaluating ML-based function predictions provided by bioinformatics researchers (Rodríguez-López *et al.* 2023), reviewing rule-based predictions from UniProtKB ARBA pipeline (UniProt Consortium 2021), and reviewing ontology definitions generated by LLMs (DRAGON-AI; Toro *et al.* 2023).

### Curating specificity

Finally, we will continue to build richer, more comprehensive connections among curated data of all types. Notably, we are converting *S. pombe* GO annotations to follow the GO Causal Activity Modeling ("GO-CAM") principles established by the GOC (Thomas *et al.* 2019). Manually curated *S. pombe* GO annotations already include a rich set of "annotation extensions" that capture reaction substrates, effector–target connections, and regulatory effects (Huntley *et al.* 2014). PomBase has already curated 3579 genes-to-gene connections for 602 genes and 5245 molecular function to biological process connections for 2372 genes. Converting these extended annotations to GO-CAM models will improve PomBase's representation of how protein activities are connected into pathways and how pathways are interconnected. To make these connections accessible to PomBase users, pathway diagrams generated using GO-CAM data will be added to the gene page summary views.

Expertly curated data are critical to the efficient and successful application of model species as research tools (Oliver *et al.* 2016; Bellen *et al.* 2021; Wood *et al.* 2022). Despite the widely acknowledged dependence of the international biological research community on biological knowledgebases, most resources face continual challenges communicating their value to funders. We hope to continue to navigate these challenges and are committed to maintaining PomBase as a reliable source of comprehensive, quality-controlled, curated knowledge derived from both large- and small-scale *S. pombe* experiments. In particular, to improve sustainability, a major aim of our next funding cycle is to integrate PomBase data into the Alliance of Genome Resources. We will also continue to collaborate closely with our user community and other resources and incorporate their valuable research and feedback to shape our strategic direction. This virtuous cycle of research providing knowledge to the MODs in turn empowers researchers to pose innovative questions that drive knowledge advancement and expedite discoveries.

## Data availability

All PomBase code is available under an open-source license from the PomBase GitHub organization (https://github.com/pombase), where each major aspect of the project—including curation, website, Chado database, FYPO, and Canto—has a dedicated repository. In addition to the web page displays and query result downloads described above, PomBase data can be downloaded in bulk from the website (see https://www.pombase.org/datasets). Available downloads include nightly database dumps and monthly releases of the entire Chado curation database and a range of specific curated data sets, including GO annotations, single-allele phenotypes, protein modifications, high-confidence physical interactions, and manually curated ortholog lists.

## Acknowledgments

PomBase thanks the fission yeast community for contributing data sets and literature curation and for providing valuable feedback on the website; Juan Mata (PI) and Jürg Bähler (Co-PI) and our advisory board for their continual support; Antonia Lock for constructive comments on the manuscript; Nico Matentzoglu (Monarch Initiative) for assistance with deploying the ODK to manage FYPO development; Daniela Raciti, Karen Yook, and the microPublications team for providing the *S. pombe* microPublications platform and ongoing support; Charlie Hoffman, Sarah Sabatinos, and Sarah Lambert for supporting our *microPublication* editorial process; Pascale Gaudet and Marc Feuermann for implementing our recommendations for improving GO annotations and ontology content; Snezka Oliferenko for overseeing the community submissions for JaponicusDB; Thomas Schalch for input in structure displays; James Seager (Rothamsted Research) for contributions to Canto; Jaqueline Hayles for contributions to PomBase; and Jon Hollis, Matthew Fairbairn, and team at the Babraham Institute for hosting and supporting the PomBase website.

## Funding

This research was funded in whole by the Wellcome Trust (grant number 218236/Z/19/Z to Juan Mata). For the purpose of open access, the author has applied a CC BY public copyright license to any author accepted manuscript version arising from this submission.

## Conflicts of interest

The authors declare no conflicts of interest.

## Author contributions

V.W. and K.M.R. wrote the manuscript. K.M.R. developed the website, Canto software, and maintained the database. V.W. and M.L.-R. performed data and literature curation. M.L.-R. wrote analysis scripts and reviewed the manuscript. All the authors commented on the manuscript draft.

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
