## [Peer Review File · Genetics]

PomBase, A Global Core Biodata Resource: Growth, Collaboration and Sustainability

Kim Rutherford, Valerie Wood, and Manuel Lera-Ramírez

NOTE: The reviews and decision letters are unedited and appear as submitted by the reviewers.

In extremely rare instances and as determined by a Senior Editor or the EIC, portions of a review may be redacted. If a review is signed, the reviewer has agreed to no longer remain anonymous.

The review history appears in chronological order.

Review Timeline:

Submission Date:	2023-11-28
Editorial Decision:	2023-12-21
Revision Received:	2024-01-03
Accepted:	2024-01-13

December 21, 2023

RE: GENETICS-2023-306680

Dear Dr. Rutherford:

I am pleased to accept your manuscript entitled "PomBase, A Global Core Biodata Resource: Growth, Collaboration and Sustainability" for publication in GENETICS, pending minor revision.

Please submit your revision along with a response to the reviewers' concerns and suggestions, which can be viewed at the bottom of this email. Most important are [INDICATE THE REVISIONS THAT ARE NECESSARY FOR THE MANUSCRIPT TO BE ACCEPTABLE]. I expect this can be done within 30 days.

Upon resubmission, please include:

1. A clean version of your manuscript;
2. A marked version of your manuscript in which you highlight significant revisions carried out in response to the major points raised by the editor/reviewers (track changes is acceptable if preferred);
3. A detailed response to the editor's/reviewers' comments and to the concerns listed above. Please reference line numbers in this response to aid the editors.

Additionally, please ensure that your revision is formatted for GENETICS: <https://academic.oup.com/genetics/pages/general-instructions>.

Follow this link to submit the revised manuscript: Link Not Available

Thank you for submitting your research to Genetics.

Sincerely,

Judith Blake
Associate Editor
GENETICS

Approved by:
Paul Sternberg
Senior Editor
GENETICS

Reviewer comments:

Reviewer #1 (Comments for the Authors (Required)):

PomBase, the comprehensive database for the fission yeast *Schizosaccharomyces pombe*, is a leader in model organism biocuration. In addition to providing curated data of the highest quality, PomBase continually develops, and shares, tools that improve the quality and quantity of annotations not only for *S. pombe* but for other species, as well. The extent to which PomBase collaborates with other biocuration groups and resources makes them an exemplar of how to build and sustain an integrated biocuration ecosystem.

As with previous database updates, the current manuscript, PomBase, A Global Core Biodata Resource: Growth, Collaboration and Sustainability, provides PomBase users with valuable information on the latest site features and other biological databases with inspiration for improving their own resources. In addition, this paper newly highlights the designation of PomBase as a GCBR by the Global Biodata Coalition, a distinction currently given to only 52 resources or knowledgebases.

I have some minor suggestions for improving the manuscript:

1. The GBC is very important to the resources it is supporting, but I suspect is not nearly as well known amongst

the bench scientists that might otherwise read Genetics. Given that, I think a summary table that lists the GBC indicators and how PomBase strives to implement those indicators would be a helpful addition to the textual description in the Introduction. You might even highlight, in color, the indicators that are the main focus of your update as described in the last paragraph of the Introduction.

2. The second paragraph of the Introduction describes PomBase as a MOD (model organism database) or MOK (model organism knowledgebase). Since that is the only mention of MOK in the text, I'd consider leaving it out since without any further explanation of the distinctions between the two, MOK doesn't add much and is potentially confusing. Alternatively, you could explain MOK and use that throughout.

3. In the Figure 1 legend and in the text you refer to 'grouping terms', 'grouping classes', and 'subclasses'. For consistency, you might want to select and use just one.

4. For the "conserved unknown" genes, are there any functions known in the other species outside the Schizosaccharomyces clade that just aren't applicable to *S. pombe*?

5. When discussing coverage, you state that, "...at PomBase (and all other MODs), much of the valuable experimental detail....remains to be captured." That's a strong general statement to make wrt "all other MODs" so unless you can cite the source of that information, I'd temper it a bit by saying "...at PomBase (and likely other MODs)...." or something similar.

6. In the section on Collaboration, you say that "MODs curate and store data using universal standard formats....". If only this were true! You might want to give an example here to explain what you mean by a 'universal standard format'.

7. The 'Mouseover provides features details' text in the Figure 2 legend is highlighted in hot pink.

8. Just in case readers don't know, I'd indicate what RCSB stands for the first time it appears in the text: RCSB (Research Collaboratory for Structural Bioinformatics).

9. The concept of FAIR biological data makes an appearance quite late in the paper under the community collaboration section, but it really applies to PomBase generally, so it would make sense to introduce this earlier.

10. This second sentence in the section on Quality control contributions wrt research progress seems quite broad and is not referenced by any analyses. If there is a reference to support this statement, it would be great to include it; otherwise, I'd leave it out.

11. In the first sentence of the first section of the Conclusions section, I'd suggest saying "Biological databases may give the appearance of being disparate...."

12. It may not be appropriate to comment in this article, but given the likely readership of Genetics, I imagine that there may be some questions as to if/how PomBase interacts with the Alliance of Genome Resources and what the future of that might look like.

13. There are a few typos to correct:

a) In the section on slims, "We use the biological process aspect of the Gene Ontology (GO)..."

b) In the ACKnowledgements, "...Daniela Raciti, Karen Yook and the microPublication Biology team..." this currently says "...and the of the microPublications team...."

14. Please double-check that all of the other resources cited in the paper, e.g. FungiDB, have an appropriate citation.

Reviewer #2 (Comments for the Authors (Required)):

The present manuscript describes recent curation and development activities at Pombase, the model organism database for *S.pombe*. The manuscript is concisely written, and is informative for both research scientists and database professionals. There are no major flaws. There are several things that could be improved, all of which should be straightforward to address.

1. The figure legends should be stand-alone, but are missing context, URLs, spelling out of acronyms, etc. They sound insular, like they are written for staff, instead of for a broader audience.

2. Authors should be consistent with British or American spelling. I see both in the manuscript.

3. "sprint project" is Agile jargon, and makes the manuscript sound insular.
4. Should provide URL for Canto in the Canto paragraph.
5. Consistency of spelling: knowledgebase or knowledge base. Both versions are present.
6. Need to spell out/define other groups so far identified only by acronyms: HGNC, DIOPT, SGD, etc.
7. Mondo should be MonDO.
8. Batten's disease should be 'Batten disease'.
9. Should provide URL or publication (?) for ISB

Do the authors' conclusions provide new insights into a biological process? If "No", please explain in comments to the editor and/or author.

No, but this question is N/A, since this manuscript describes a knowledgebase.

Reviewer #3 (Comments for the Authors (Required)):

Dear authors,

I have reviewed your manuscript entitled "PomBase, A Global Core Biodata Resource: Growth, Collaboration and Sustainability". I am very impressed by the quality and quantity of the work that PomBase is achieving with the limited resources available.

I have a small number of very minor comments:

- Line 31, when "RNA processing" is mentioned, I was expecting a reference, to be consistent with the rest of the processes enumerated in this paragraph
- Line 40, could you be more specific about the "usual search tools" (or remove that part of the sentence)?
- Line 197, "newly published novel gene characterization" -> "novel" could probably be removed to make the sentence less redundant
- Line 271 "Nov 23" -> should be changed to November 2023 to be consistent with how other dates are written elsewhere
- Line 293, "figure 1c" should be "figure 2c"
- Line 341, change "modifications" to "protein modifications" to be more explicit
- Line 451, "November 23" should be changed to November 2023 to be consistent with how other dates are written elsewhere
- Line 523 "GBC GBCR indicators"; in line 19, this is written as "GBCR indicators"; change one or the other for consistency
- Line 566: add link to journals to clarify what you are referring to
- Line 567: the cdc15 example is a bit out of place; either remove, or add a bit of an explanation as to what this gene exemplifies here

Best regards.

Judy Blake

Associate Editor comments:

Reviewer comments with replies in **bold**:

Reviewer #1 (Comments for the Authors (Required)):

1. The GBC is very important to the resources it is supporting, but I suspect is not nearly as well known amongst the bench scientists that might otherwise read Genetics. Given that, I think a summary table that lists the GBC indicators and how PomBase strives to implement those indicators would be a helpful addition to the textual description in the Introduction. You might even highlight, in color, the indicators that are the main focus of your update as described in the last paragraph of the Introduction.

We appreciate the suggestion, but since there are 52 indicators we feel that such a table would be a little out of scope for the manuscript. Although we refer to the indicators, we only do so loosely to emphasize that it is difficult to metricise indicators related to scientific quality, rather than to be a comprehensive assessment of the GBCR Indicators. Instead we added “(see <https://zenodo.org/records/7468719> for a full list of indicators)”

2. The second paragraph of the Introduction describes PomBase as a MOD (model organism database) or MOK (model organism knowledgebase). Since that is the only mention of MOK in the text, I'd consider leaving it out since without any further explanation of the distinctions between the two, MOK doesn't add much and is potentially confusing. Alternatively, you could explain MOK and use that throughout.

Agreed and deleted.

3. In the Figure 1 legend and in the text you refer to 'grouping terms', 'grouping classes', and 'subclasses'. For consistency, you might want to select and use just one.

We now use “classes”

4. For the "conserved unknown" genes, are there any functions known in the other species outside the Schizosaccharomyces clade that just aren't applicable to *S. pombe*?

There are biological processes assigned for a small number of the unknowns, but we believe that for most these are not the direct role, but are phenotypes and readouts (the actual process is causally upstream). Many of the unknowns have molecular functions assigned but have no biological process as our schema is based on role, not activity as described in the text. In other cases, two different model species will have conflicting annotations, supporting the idea that these do not describe the true role. In other cases, the experimental data used for the annotation is very weak. In cases where there is tentative functional information, we add this to the product line “implicated in”. Others where a role is captured but is obscure (for example complex I subunits which are

conserved in yeast are described in other species as respiratory chain, but yeast do not have complex I, it is likely these peripheral subunits are not directly involved in electron transport, but have a more general role in iron-sulfur cluster insertion). There are a couple of exceptions where there is a known role in metazoa that does not apply to yeast (e.g. COMT methyltransferase), but for most of the 600 unknowns the role is either unknown, speculative, very poorly characterized or conflicting.

5. When discussing coverage, you state that, "...at PomBase (and all other MODs), much of the valuable experimental detail.....remains to be captured." That's a strong general statement to make wrt "all other MODs" so unless you can cite the source of that information, I'd temper it a bit by saying "....at PomBase (and likely other MODs)...." or something similar.

We changed as suggested.

6. In the section on Collaboration, you say that "MODs curate and store data using universal standard formats....". If only this were true! You might want to give an example here to explain what you mean by a 'universal standard format'.

**Good point! We changed the existing text to
"When Pombase requires new tools and visualizations for curated or imported data we always evaluate and reuse existing software before developing new tools in-house."**

7. The 'Mouseover provides features details' text in the Figure 2 legend is highlighted in hot pink.

Now fixed

8. Just in case readers don't know, I'd indicate what RCSB stands for the first time it appears in the text: RCSB (Research Collaboratory for Structural Bioinformatics).

Added

9. The concept of FAIR biological data makes an appearance quite late in the paper under the community collaboration section, but it really applies to PomBase generally, so it would make sense to introduce this earlier.

We added

PomBase aims to be a fully FAIR (Findable, Accessible, Interoperable and Reusable) compliant resource (cite)

to the introduction

10. This second sentence in the section on Quality control contributions wrt research progress seems quite broad and is not referenced by any analyses. If there is a reference to support this statement, it would be great to include it; otherwise, I'd leave it out.

We decided to keep this one because we have many individual examples supported by curation tracker tickets (I would classify as common knowledge the outdated science at the level of individual annotations which are removed, especially over-interpreted phenotypes). We have added the citation doi:10.1098/rsob.200149 which identifies outdated biological knowledge as a source of misannotation.

11. In the first sentence of the first section of the Conclusions section, I'd suggest saying "Biological databases may give the appearance of being disparate....."

Added

12. It may not be appropriate to comment in this article, but given the likely readership of Genetics, I imagine that there may be some questions as to if/how PomBase interacts with the Alliance of Genome Resources and what the future of that might look like.

In particular, to improve sustainability, a major aim of our next funding cycle is to integrate PomBase data into the Alliance of Genome Resources

13. There are a few typos to correct:

a) In the section on slims, "We use the biological process aspect of the Gene Ontology (GO)..."

b) In the ACKnowledgements, "...Daniela Raciti, Karen Yook and the microPublication Biology team..." this currently says "...and the of the microPublications team...."

Fixed

14. Please double-check that all of the other resources cited in the paper, e.g. FungiDB, have an appropriate citation.

We cited all of the resources that we use data or software from, or work with in a specific way. We did not think it warranted a citation for 2nd and third-party resources that host the data we produce via another resource (i.e from GO), although we mentioned their reuse of the data.

Reviewer #2 (Comments for the Authors (Required)):

1. The figure legends should be stand-alone, but are missing context, URLs, spelling out of acronyms, etc. They sound insular, like they are written for staff, instead of for a broader audience.

We have added context for all legends, full acronyms

2. Authors should be consistent with British or American spelling. I see both in the manuscript.

We have changed the remaining British spellings to American spelling

3. "sprint project" is Agile jargon, and makes the manuscript sound Insular.

Removed

4. Should provide URL for Canto in the Canto paragraph.

Added

5. Consistency of spelling: knowledgebase or knowledge base. Both versions are present.

Fixed

6. Need to spell out/define other groups so far identified only by acronyms: HGNC, DIOPT, SGD, etc.

Added

7. Mondo should be MonDO.

Mondo should be written “Mondo” but we corrected to “Monarch disease ontology (Mondo)”

8. Batten's disease should be 'Batten disease'.

Fixed

9. Should provide URL or publication (?) for ISB

Added URL

Do the authors' conclusions provide new insights into a biological process? If "No", please explain in comments to the editor and/or author.

No, but this question is N/A, since this manuscript describes a knowledgebase.

Reviewer #3 (Comments for the Authors (Required)):

- Line 31, when "RNA processing" is mentioned, I was expecting a reference, to be consistent with the rest of the processes enumerated in this paragraph

added

- Line 40, could you be more specific about the "usual search tools" (or remove that part of the sentence)?

Updated text

- Line 197, "newly published novel gene characterization" -> "novel" could probably be removed to make the sentence less redundant

Updated

- Line 271 "Nov 23" -> should be changed to November 2023 to be consistent with how other dates are written elsewhere

Changed

- Line 293, "figure 1c" should be "figure 2c"

Fixed

- Line 341, change "modifications" to "protein modifications" to be more explicit

Changed

- Line 451, "November 23" should be changed to November 2023 to be consistent with how other dates are written elsewhere

Changed

- Line 523 "GBC GBCR indicators"; in line 19, this is written as "GBCR indicators"; change one or the other for consistency

Changed

- Line 566: add link to journals to clarify what you are referring to

We have added a little more context.

As a proof of principle, Genetics, G3 and microPublications now provide deep links from publications into PomBase for gene mentions (e.g. *cdc15*), by using a lexicon of PomBase entities.

There is currently no publication or public-facing documentation describing this procedure as it is relatively new.

- Line 567: the *cdc15* example is a bit out of place; either remove, or add a bit of an explanation as to what this gene exemplifies here

We put this in as an example because this will be an example of a hyperlinked entity

January 8, 2024

RE: GENETICS-2023-306680R1

Mr. Kim M. Rutherford
University of Cambridge
Biochemistry
Tennis Court Rd.
Cambridge
United Kingdom

Dear Dr. Rutherford:

Congratulations! We are delighted to inform you that your manuscript entitled "PomBase, A Global Core Biodata Resource: Growth, Collaboration and Sustainability" is acceptable for publication in GENETICS. Many thanks for submitting your research to the journal.

To Proceed to Production:

Add oupsupport@scipris.com and genetics.oup@novatechset.com (or the domains @scipris.com and @novatechset.com) to your email program's "safe senders" list. You will be contacted by both at various points during the production process.

1. Format your article according to GENETICS style, as discussed at <https://academic.oup.com/genetics/pages/general-instructions>. Ensure that you comply with data and community resource citation guidelines (<https://academic.oup.com/genetics/pages/general-instructions#Data-Policy>).
2. Upload your final files at <https://genetics.msubmit.net>.
3. Your currently-accepted manuscript (unedited, as submitted, reviewed, and accepted) will be published at GENETICS and deposited into PubMed as an Advance Access article. Notify sourcefiles@thegsajournals.org before signing your license if you do not wish to publish your article via Advance Access.
4. We invite you to submit an original color figure related to your paper for consideration as cover art. Please email your submission to the editorial office or upload it with your final files. You can submit a small-sized image for evaluation, and if selected, the final image must be a TIFF file 2513px wide by 3263px high (8.375 by 10.875 inches; resolution of 600ppi). Please avoid graphs and small type.
5. After files are sent to Oxford University Press we use SciPris to manage article licensing and payment. If you do not have a SciPris account, you will receive an email from no-reply@scipris.com to sign up to use Oxford University Press' author portal. After logging in, follow the online instructions to sign your licence and arrange any payment due.

If you have any questions or encounter any problems while uploading your accepted manuscript files, please email the editorial office at sourcefiles@thegsajournals.org.

Sincerely,

Judith Blake
Associate Editor
GENETICS

Approved by:
Paul Sternberg
Senior Editor
GENETICS